# Residual physiological abnormalities after pulmonary endarterectomy and balloon pulmonary angioplasty in CTEPH

Rita Calé[1]*, Filipa Ferreira[1], Mariana Martinho[1], Sofia Alegria[1], Débora Repolho[1], Ana Rita Pereira[1], João Luz[1], Tiago Lobão[1], Patrícia Araújo[1], Sílvia Vitorino[1], Hélder Pereira[1,2], Daniel Caldeira[2,3,4]

1 Serviço de Cardiologia, Hospital Garcia de Orta – ULS Almada e Seixal, Almada, Portugal, 2 Centro Cardiovascular da Universidade de Lisboa-CCUL (CCUL@RISE), CAML, Faculdade de Medicina, Universidade de Lisboa, Lisboa, Portugal, 3 Serviço de Cardiologia, Departamento do Coração e Vasos, Hospital Universitário de Santa Maria - ULS Santa Maria, Lisboa, Portugal, 4 Centro de Estudos de Medicina Baseada na Evidência (CEMBE), Faculdade de Medicina, Universidade de Lisboa, Lisboa, Portugal

* ritacale@hotmail.com

## Abstract

### Introduction

Pulmonary endarterectomy (PEA) is the first-line treatment for chronic thromboembolic pulmonary hypertension (CTEPH), while balloon pulmonary angioplasty (BPA) is an established alternative for inoperable patients. Although both interventions improve resting pulmonary hemodynamics, the extent of long-term physiological recovery during exercise and the persistence of functional limitations remain incompletely characterized.

### Methods

Prospective single-center registry (2017–2023) including 14 patients completing BPA (71 sessions) and 15 undergoing PEA, with median follow-up of 50 months (IQR 36–61). Clinical assessment included resting hemodynamics, invasive exercise right heart catheterization to derive the exercise slope of the mean pulmonary arterial pressure to cardiac output relashionship (mPAP/CO slope), and health-related quality of life (HRQOL) evaluated using the SF-36 questionnaire. Analyses were descriptive and focused on within-pathway changes over time.

### Results

Both BPA and PEA significantly reduced mPAP ($44.8 \pm 12.4 \rightarrow 26.1 \pm 9.3$ mmHg; $42.1 \pm 12.9 \rightarrow 22.6 \pm 5.4$ mmHg, both $p < 0.001$) and pulmonary vascular resistance ($9.8 \pm 4.6 \rightarrow 3.0 \pm 1.3$ WU; $9.0 \pm 5.4 \rightarrow 2.9 \pm 1.9$ WU, both $p < 0.001$) at long term follow-up. Despite sustained improvements in resting hemodynamics, abnormal

**Data availability statement:** Public availability of the full dataset will not be possible because it contains single-center clinical data regarding a relatively uncommon pathology, which makes true anonymization unattainable without seriously compromising scientific utility. Even after removal of direct identifiers, the combination of center-specific characteristics, detailed clinical variables, and the rarity of the condition creates a substantial risk of deductive disclosure of individual participants, particularly within the local community or professional networks. Current best practices and international guidance on sharing human participant data emphasize that data sharing should never compromise participant privacy and must remain fully aligned with the scope of informed consent and applicable ethical and legal frameworks. In this context, unrestricted public deposition of the complete dataset would conflict with these core principles of confidentiality and respect for persons. Therefore, and in accordance with policies that explicitly recognize ethical restrictions as a legitimate limitation to open data, access to the dataset can only be offered under controlled conditions, via an appropriate institutional or ethics committee–governed mechanism, to qualified researchers who present a scientifically sound proposal and commit to robust safeguards for privacy and data security. Data are available from the Hospital Garcia de Orta for researchers who meet the criteria for access to confidential data. Requests for data access can be directed to commissao.etica@ulsas.min-saude.pt.

**Funding:** The author(s) received no specific funding for this work.

**Competing interests:** The authors have declared that no competing interests exist.

exercise pulmonary vascular responses persisted, with mean mPAP/CO slopes of 7.0±5.6 mmHg/L/min after BPA and 4.0±2.3 mmHg/L/min after PEA. Physical HRQOL remained impaired at long-term follow-up, with Physical Component Summary (PCS) scores below population norms in both pathways (44.4±12.7 after BPA and 44.5±7.3 after PEA).

## Conclusion

BPA and PEA provide durable improvements in resting pulmonary hemodynamics; however, incomplete physiological recovery is common, with persistent exercise abnormalities and reduced physical quality of life at long-term follow-up.

---

## Introduction

Chronic thromboembolic pulmonary hypertension (CTEPH) is a chronic, progressive disease characterized by obstruction of the pulmonary arterial vasculature by organized thrombotic material. These changes trigger a fibrotic response, intimal thickening, and progressive vascular remodelling, including the formation of plexiform lesions, which collectively contribute to increased pulmonary vascular resistance (PVR) and pressure. The resulting hemodynamic burden leads to severe pulmonary hypertension, right ventricular overload, progressive right heart failure, and ultimately death [1–3]. When treated solely with medical therapy, CTEPH is associated with a poor prognosis, with 1-, 3-, and 5-year survival rates of 90.2%, 78.4%, and 64.5%, respectively [4].

According to the 2022 European Society of Cardiology (ESC)/European Respiratory Society (ERS) guidelines on the diagnosis and treatment of pulmonary hypertension, the management of CTEPH is multimodal, combining pulmonary endarterectomy (PEA), balloon pulmonary angioplasty (BPA), and pulmonary vasodilator therapy, when indicated. This integrated approach allows for targeted intervention at different levels of the pulmonary arterial tree, from major pulmonary vessels to the microcirculation, aiming to optimize hemodynamic outcomes and clinical prognosis [5].

PEA is the treatment of choice for eligible patients and, when performed in experienced centers, is associated with in-hospital mortality rates below 5% and provides significant hemodynamic and functional improvements with good long-term survival [6,7]. However, up to 37% of patients are deemed inoperable [8–10], and residual or recurrent pulmonary hypertension occurs in as many as 50% of patients undergoing PEA surgery, with consequent impact on morbidity and mortality [11]. In these settings, BPA is currently regarded as a valuable therapeutic option within the CTEPH management algorithm [5,12].

Both PEA [6,13] and BPA [14–17] can result in substantial and sustained improvements in resting pulmonary hemodynamics and functional capacity. Nevertheless, a significant proportion of patients continue to experience residual symptoms and functional limitations following treatment [11]. Moreover, data on long-term health-related

quality of life and hemodynamic responses during exercise in this population remain scarce. Understanding these outcomes is essential to determine whether invasive therapies lead to complete clinical and physiological resolution of the disease.

## Objective

This study aimed to describe long-term health-related quality of life and exercise-induced hemodynamic responses in patients with CTEPH treated with PEA or BPA, in order to explore the extent of physiological recovery achieved after each treatment pathway.

## Methods

### Study population and ethics

This prospective observational cohort study included consecutive patients with inoperable CTEPH who underwent BPA at Hospital Garcia de Orta (Almada, Portugal) between 2017—when the BPA program was established—and December 2023 (**BPA pathway**). During the same period, patients undergoing pulmonary endarterectomy (**PEA pathway**) at the same center were also included and analyzed as a separate treatment pathway. To preserve interpretability given distinct treatment trajectories, patients managed with a hybrid strategy combining PEA and BPA were not included in the primary analysis and are described separately in the Supplementary Material.

A prospective long-term follow-up was conducted between January 2021 and August 2025, during which all surviving patients were contacted and provided written informed consent for participation and use of their clinical data. Clinical data were accessed for research purposes on 30 January 2021. The dataset included identifiable information required for accurate patient follow-up; only authorized members of the research team at Hospital Garcia de Orta had access to these data. All procedures complied with institutional and ethical standards, and the study protocol was approved by the Ethics Committee of Hospital Garcia de Orta (approval no. 98/2020) in accordance with the Declaration of Helsinki.

The classification of CTEPH at the time of patient inclusion was based on the most recent hemodynamic definition of pulmonary hypertension, as proposed by the 6th World Symposium on Pulmonary Hypertension [18], adopted in subsequent guidelines [5], and maintained in the 7th World Symposium [19]. Specifically, CTEPH was defined by a mPAP > 20 mmHg, a pulmonary capillary wedge pressure (PCWP) ≤15 mmHg, and a pulmonary vascular resistance (PVR) > 2 Wood units, in the presence of mismatched perfusion defects on imaging and chronic thromboembolic lesions on pulmonary angiography [5]. Patients who underwent BPA for chronic thromboembolic pulmonary disease but did not meet the diagnostic criteria for CTEPH under this updated definition were excluded from the study.

Participants underwent a complete diagnostic study including clinical history and assessment of comorbidities, pulmonary ventilation-perfusion scan, computed tomography angiography of the thorax with biplane reconstructions, digital subtraction pulmonary angiography, and right heart catheterization.

All patients with CTEPH were evaluated by a multidisciplinary team from a surgical center with expertise in PEA. Based on this evaluation, patients were recommended for surgical treatment, BPA or medical therapy. In some cases, pulmonary vasodilator therapy is combined with subsequent surgical or percutaneous intervention as part of a multimodal treatment approach.

### Assessment before and after intervention (BPA or PEA)

All patients underwent a comprehensive clinical assessment at the time of CTEPH diagnosis (baseline). In the BPA pathway, an additional pre-intervention assessment was performed immediately before the first BPA session, following a period of optimization with pulmonary vasodilator therapy. Follow-up evaluations were conducted six months after surgery or after the final BPA session, and at long-term follow-up (more than two years post-intervention). The clinical assessment

included World Health Organization (WHO) functional class, six-minute walk distance (6MWD), serum N-terminal pro-brain natriuretic peptide (NT-proBNP) levels, arterial blood gas analysis, and resting right heart catheterization. Hemodynamic parameters measured directly included right atrial pressure (RAP); systolic, diastolic and mean pulmonary arterial pressures (sPAP, dPAP, mPAP) and PCWP. Hemodynamic parameters measured indirectly were PVR, and cardiac output, the latter determined by the thermodilution method. Right heart catheterization under exercise conditions was performed at long-term follow-up.

## Health-related quality of life (HRQOL) assessment

Health-related quality of life was assessed in both the surgical and percutaneous intervention groups at long-term follow-up using the RAND 36-Item Short Form Health Survey, version 1 (SF-36; https://orthotoolkit.com/sf-36/). This validated questionnaire [20,21] evaluates eight domains of physical and mental health, allowing a comprehensive assessment of patients' perceived well-being after treatment: physical functioning (PF), role physical (RP), bodily pain (BP), general health (GH), vitality (VT), social functioning (SF), role emotional (RE), and mental health (MH). The SF-36 scores range from 0 to 100, with higher scores indicating better HRQOL. Each domain (raw score) was calculated and normalized against a norm-based U.S.population score, with scores scaled to a national mean of 50 and a standard deviation of 10 (normalized score). A score below 50 indicated worse score than the normative general population, while every 10 points represented one SD.

Two summary scores were calculated from the 8 subscales: physical component summary (PCS) and mental component summary (MCS) [20].

The version used in this study was a validated translation of the original English questionnaire [22].

## Exercise right heart catheterization protocol

Exercise right heart catheterization (RHC) was performed using a supine cycle ergometer (*Lode Angio Imaging*, Lode BV, Groningen, The Netherlands). A Swan-Ganz catheter was inserted via the brachial vein under local anesthesia.

Hemodynamic parameters, including pulmonary arterial pressures, pulmonary arterial wedge pressure, and cardiac output, were obtained at rest and during a symptom-limited, stepwise incremental cycling exercise protocol, starting at 10 watts, with workload increments of 10 watts every 3 minutes until the patient developed limiting symptoms or completed 15 minutes of exercise (reaching a maximum of 50 watts), while maintaining the supine position throughout. Hemodynamic data were recorded at each workload and 5 minutes after exercise cessation. Cardiac output was measured using thermodilution method. Mixed venous oxygen saturation ($SvO_2$) was measured by blood gas analysis from pulmonary artery samples at rest and at peak exercise, defined as the final workload achieved. Maximal effort was defined by an $SvO_2 < 30\%$ at peak exercise [23].

The pulmonary vascular response to exercise was assessed by calculating the slope of the mean pulmonary arterial pressure to cardiac output relationship (mPAP/CO slope). This parameter was obtained by plotting the change in mean pulmonary arterial pressure (ΔmPAP) against the corresponding change in cardiac output (ΔCO) between rest and peak exercise. The slope was expressed in $mmHg \cdot L^{-1} \cdot min^{-1}$. A value greater than 3 $mmHg \cdot L^{-1} \cdot min^{-1}$ was considered indicative of an abnormal pulmonary hemodynamic response to exercise, as suggested by current guidelines [5].

To assess the post-capillary contribution to exercise hemodynamics, the pulmonary capillary wedge pressure to cardiac output relationship (PCWP/CO slope) was also calculated. This was determined by plotting the change in pulmonary capillary wedge pressure (ΔPCWP) against the change in cardiac output (ΔCO) from rest to peak exercise. The slope was expressed in $mmHg \cdot L^{-1} \cdot min^{-1}$. A PCWP/CO slope greater than 2 $mmHg \cdot L^{-1} \cdot min^{-1}$ was considered indicative of an abnormal post-capillary response to exercise, suggestive of left heart involvement [5].

### Balloon pulmonary angioplasty

BPA was performed according to the established institutional protocol [24].

Patients remained on anticoagulation and pulmonary vasodilator therapy throughout the treatment. All procedures were performed in conscious patients under local anesthesia, using ultrasound-guided femoral venous access. BPA was delivered in staged sessions using soft guidewires and semi-compliant balloons, with lesion complexity and hemodynamic severity guiding procedural strategy. Sessions were repeated every 3–4 weeks until predefined hemodynamic targets were achieved or all accessible lesions were treated.

All BPA procedures were performed by the same two dedicated operators. The initial cases were proctored by a high-volume international BPA operator, after which the program continued independently.

### Pulmonary endarterectomy

PEA was performed using standard techniques at experienced surgical centers with established expertise in CTEPH management, under cardiopulmonary bypass with deep hypothermia and intermittent circulatory arrest, with bilateral endarterectomy extending to segmental and subsegmental pulmonary arteries [7].

All procedures were performed at referral centers within an established collaborative network. Most were undertaken in two high-volume centres.

### Statistical analysis

Statistical analyses were performed using IBM SPSS Statistics 25.0 (IBM Corp., Armonk, NY, USA). Categorical variables were presented as counts and percentage, and continuous variables as mean ± standard deviation for normally distributed data or median and interquartile range for non-normally distributed data.

Given the observational design and the distinct treatment pathways leading to BPA or PEA, analyses were primarily descriptive and focused on within-group changes over time. Changes in continuous variables within each treatment pathway were assessed using paired Student's $t$ tests for normally distributed data or Wilcoxon signed-rank tests for non-normally distributed data. Changes in categorical variables over time were evaluated using McNemar's test.

Reductions in resting mean pulmonary arterial pressure (mPAP) and pulmonary vascular resistance (PVR) following BPA were illustrated by calculating percentage changes using both individual patient-level values (reported as the mean of individual percentage changes) and group-level mean values, to allow comparison with published registry data.

Exercise hemodynamics (mPAP/CO slope) and health-related quality of life measures (SF-36 physical and mental component summary scores) are presented descriptively for each treatment pathway.

Exploratory subgroup analyses were performed at long-term follow-up according to post-procedural resting hemodynamic status and use of pulmonary vasodilator therapy, and are presented descriptively.

All analyses were conducted with a two-sided significance level of 5%, where applicable.

## Results

### Population characteristics

Between December 2017 and December 2023, 36 patients with CTEPH were evaluated at a pulmonary hypertension reference center by a multidisciplinary team, and allocated to either a BPA or PEA treatment pathway. Seventeen patients were deemed inoperable and referred for a BPA program due to predominantly distal disease (n = 10), an unfavorable risk/benefit profile for PEA (n = 6), or surgical refusal (n = 1). Of these, 14 patients completed the BPA program and were included in the long-term analysis. Nineteen patients underwent PEA, of whom 15 were included in the long-term analysis; those requiring subsequent BPA for residual pulmonary hypertension were analyzed separately and are described in the Supplementary Material. Patient selection and exclusions are detailed in Fig 1.

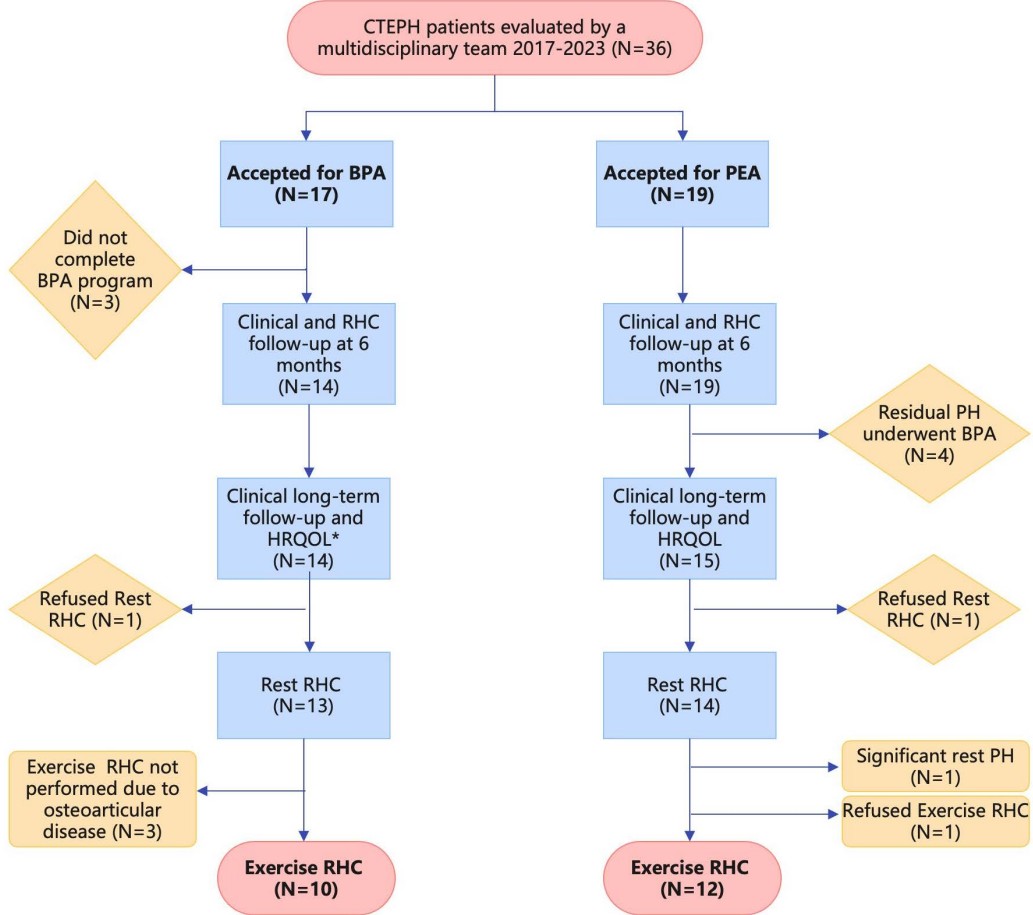

**Fig 1. Flowchart of CTEPH patients from initial evaluation to long-term follow-up, including clinical assessment, HRQOL, and resting and exercise right heart catheterization.** * One patient who died during follow-up was not available for long-term HRQOL assessment (SF-36).

The baseline characteristics of patients included in both groups are presented in Table 1.

Baseline characteristics were broadly similar across treatment pathways. Patients referred for BPA more frequently had chronic obstructive pulmonary disease and more advanced functional limitation at baseline, reflecting the influence of comorbidities and disease distribution on treatment allocation by the multidisciplinary team. In addition, after the baseline assessment, four patients in the BPA group required stabilization with prostacyclin analogs, a therapy that was not used in the surgical group. Baseline exercise capacity (6-minute walk distance), NT-proBNP levels, and resting hemodynamic severity were otherwise comparable.

In addition, a descriptive overview of patients who underwent a hybrid approach (PEA followed by BPA) is provided in Supplementary Table S1.

**Procedural characteristics**

The 14 inoperable CTEPH patients included in the BPA pathway completed a total of 71 BPA sessions between 2017 and 2023, with a mean of 5.1±1.8 sessions per patient. A total of 325 vessels and 192 segments were treated, corresponding to a mean of 23.2±10.0 vessels and 10.2±2.6 segments per patient. The majority of treated lesions were webs (65.8%), followed by subtotal (18.5%) and total occlusions (11.7%), and stenoses (4.0%).

**Table 1. Population characteristics at baseline.**

| Category | Variable | BPA (n = 14) | PEA (n = 15) |
|---|---|---|---|
| Demographics | Age (years) | 66.2±13.0 | 63.5±17.7 |
| | Age ≥ 75 years | 3 (21.4) | 4 (26.7) |
| | Female sex, n (%) | 8 (57.1) | 9 (60.0) |
| | BMI (kg/m²) | 28.0±4.7 | 28.5±5.4 |
| Medical History | Previous VTE, n (%) | 10 (71.4) | 12 (80.0) |
| | Hypertension, n (%) | 10 (71.4) | 6 (40.0) |
| | Diabetes mellitus, n (%) | 3 (21.4) | 2 (13.3) |
| | Thrombophilic disorder, n (%) | 1 (7.1) | 2 (13.3) |
| | COPD, n (%) | 4 (28.6) | 0 (0) |
| | History of cancer, n (%) | 3 (21.4) | 1 (6.7) |
| Functional Capacity | WHO FC, n (%) | | |
| | Class I | 0 | 0 |
| | Class II | 5 (35.7) | 4 (26.7) |
| | Class III | 5 (35.7) | 10 (66.7) |
| | Class IV | 4 (28.6) | 1 (6.7) |
| | 6MWD (m) | 275 (210-390) | 270 (153-437) |
| Laboratory | Creatinine (mg/dL) | 1.0±0.3 | 1.0±0.2 |
| | NT-proBNP (ng/L) | 570 (389-1678) | 340 (89-1649) |
| | $PaO_2$, mmHg | 63.8±9.9 | 66.1±12.9 |
| Medication | Vasodilator therapy, n (%) | | |
| | None | 4 (28.6) | 8 (53.3) |
| | Monotherapy | 2 (14.3) | 2 (13.3) |
| | Dual therapy | 5 (35.7) | 4 (26.7) |
| | Triple therapy | 3 (21.4) | 1 (6.7) |
| | sGC stimulator, n (%) | 7 (50.0) | 3 (30.0) |
| | ERA, n (%) | 7 (50.0) | 5 (33.3) |
| | PDE-5 inhibitor, n (%) | 3 (21.4) | 5 (33.3) |
| | Prostacycline analogs, n (%) | 4 (28.6) | 0 |
| | Vitamin K antagonists, n (%) | 5 (35.7) | 2 (13.3) |
| | DOAC, n (%) | 9 (64.3) | 13 (86.7) |
| | Long-term oxygen therapy, n (%) | 6 (42.9) | 2 (13.3) |
| Echocardiography | RA volume/BSA (ml/m²) | 43.5±27.4 | 31.7±20.6 |
| | RV diastolic area (mm²) | 29.0±9.9 | 21.7±6.4 |
| | RV systolic area (mm²) | 20.4±9.2 | 16.8±7.4 |
| | RV FAC (%) | 29.1±11.9 | 31.7±9.9 |
| | TAPSE (mm) | 18.6±5.1 | 18.9±3.9 |
| | RV S´(cm/s) | 10.7±3.0 | 10.3±2.2 |
| | LV diastolic EI | 1.3±0.4 | 1.2±0.2 |
| | LV systolic EI | 1.6±0.8 | 1.4±0.4 |
| | sPAP (mmHg) | 74.2±29.7 | 78.3±30.7 |
| | TAPSE/sPAP ratio (mm/mmHg) | 0.32±0.19 | 0.31±0.20 |

*(Continued)*

**Table 1.** (Continued)

| Category | Variable | BPA (n = 14) | PEA (n = 15) |
|---|---|---|---|
| Haemodynamics | RAP (mmHg) | 7.6 ± 4.6 | 6.7 ± 3.3 |
| | sPAP (mmHg) | 72.2 ± 21.9 | 76.1 ± 25.9 |
| | dPAP (mmHg) | 26.7 ± 10.2 | 23.7 ± 7.8 |
| | mPAP (mmHg) | 43.6 ± 12.7 | 42.7 ± 12.6 |
| | PCWP (mmHg) | 9.9 ± 2.5 | 11.3 ± 4.8 |
| | SvO$_2$ (%) | 62.3 ± 9.2 | 67.3 ± 6.7 |
| | CO (L/min) | 4.2 ± 1.2 | 4.2 ± 1.3 |
| | CI (L/min/m²) | 2.2 ± 0.5 | 2.4 ± 0.6 |
| | PVR (woods units) | 9.5 ± 4.6 | 9.0 ± 5.2 |
| | PAC (mL/mmHg) | 1.4 ± 0.7 | 1.5 ± 1.0 |
| | HR (beats/min) | 76.5 ± 12.4 | 72.8 ± 9.7 |

*Baseline characteristics are presented descriptively; no statistical comparisons between groups were performed given the observational design and different treatment pathways.*

6MWD: 6-min walk distance; BMI: body mass index; BPA: balloon pulmonary angioplasty; EI – excentricity index; CO: cardiac output; COPD: chronic obstructive pulmonary disease; CI: cardiac index; DOAC: direct oral anticoagulant; dPAP: diastolic pulmonary arterial pressure; ERA: endothelin receptor antagonists; HR: heart rate; LV: left ventricle; mPAP: mean pulmonary arterial pressure; PAC: pulmonary arterial compliance; PaO2: partial pressure of oxygen; PCWP: pulmonary capillary wedge pressure; PDA: phosphodiesterase; PEA: pulmonary endarterectomy; PH: pulmonary hypertension; PVR: pulmonary vascular resistance; RA: right atrium; RV FAC: right ventricular fractional area change; SvO$_2$: mixed venous oxygen saturation; sGC: soluble guanylate cyclase; sPAP: systolic pulmonary arterial pressure; VTE: venous thromboembolism; TAPSE: tricuspid annular plane systolic excursion; WHO FC: World Heath Organization functional class.

Procedure-related complications occurred in 12.7% of sessions and were predominantly mild. These included limited vascular injury (2.8%), hemoptysis (4.2%) and mild pulmonar injury (4.2%), all managed conservatively without the need for invasive mechanical ventilation, extracorporeal membrane oxygenation, or emergency surgery. Vascular access site complications and contrast nephropathy occurred in only one session each (1.8%). No patients presented acute radiation-induced dermatitis. No major procedure-related complications were noted and there were no periprocedural deaths. One patient in the BPA group died because of cancer after 28 months of follow-up.

## Results in clinical and Rest RHC at long-term

Long-term hemodynamic assessment was available for 13 patients in the BPA pathway and 14 patients in the PEA pathway, as one patient in each group declined repeat RHC at long-term follow-up. The median time from intervention to rest RHC was 40 months (IQR 29–49) after BPA and 45 months (IQR 34–62) after PEA. Long-term clinical outcomes and hemodynamic parameters in both groups are summarized in Table 2.

Participation in cardiac rehabilitation after intervention was low (7.1% in BPA and 20.0% in PEA). At long-term follow-up, pulmonary vasodilator therapy was used mainly in the BPA pathway (71.4%), whereas only isolated use was observed after PEA (6.7%).

Functional status improved from baseline to follow-up in both treatment pathways, as illustrated in Fig 2.

Despite the absence of a significant reduction in NT-proBNP levels, echocardiography showed right ventricular reverse remodeling at long-term follow-up, with improved systolic function and RV–pulmonary arterial coupling after both BPA and PEA.

Regarding the hemodynamic parameters, there was a clear and significant improvement in terms of reduced pressures in the pulmonary artery, decreased pulmonary vascular resistance, increased cardiac output, and consequent improvement in pulmonary compliance in both interventions (Table 2).

**Table 2. Long-term changes in clinical, echocardiographic and resting hemodynamic parameters.**

| Category | Variable | BPA (n = 14) | | | PEA (n = 15) | | |
|---|---|---|---|---|---|---|---|
| | | Baseline | Follow-up | p value | Baseline | Follow-up | p value |
| Functional Capacity | WHO FC | | | 0.003 | | | <0.001 |
| | Class I | 0 | 8 (57.1) | | 0 | 12 (80.0) | |
| | Class II | 5 (35.7) | 5 (35.7) | | 4 (26.7) | 3 (20.0) | |
| | Class III | 5 (35.7) | 1 (7.1) | | 10 (66.7) | 0 | |
| | Class IV | 4 (28.6) | 0 | | 1 (6.7) | 0 | |
| | 6MWD (m) | 270 (190-360) | 360 (300-480) | 0.108 | 400 (105-465) | 375 (323-525) | 0.078 |
| Laboratory | Creatinine (µmol/L) | 1.0±0.3 | 1.0±0.3 | 0.218 | 1.0±0.2 | 1.0±0.4 | 0.670 |
| | NT-proBNP (ng/L) | 570 (389-1678) | 123 (76-512) | 0.116 | 340 (89-1649) | 136 (72-267) | 0.100 |
| Echocardiography | RA volume/BSA (ml/m²) | 43.5±27.4 | 25.3±8.9 | 0.042 | 31.7±20.6 | 23.4±7.2 | 0.103 |
| | RV diastolic area (mm²) | 29.0±9.9 | 21.2±7.4 | 0.004 | 21.7±6.4 | 19.2±3.0 | 0.175 |
| | RV systolic area (mm²) | 20.4±9.2 | 11.2±4.2 | <0.001 | 16.8±7.4 | 11.5±2.4 | 0.010 |
| | RV FAC (%) | 29.1±11.9 | 47.1±4.1 | <0.001 | 31.7±9.9 | 38.8±7.6 | 0.024 |
| | TAPSE (mm) | 18.6±5.1 | 22.0±2.8 | 0.047 | 19.4±3.5 | 17.8±2.5 | 0.036 |
| | RV S' (cm/s) | 10.7±3.0 | 14.1±2.2 | <0.001 | 10.3±2.2 | 10.5±2.5 | 0.804 |
| | LV diastolic EI | 1.3±0.5 | 1.0±0.0 | 0.063 | 1.2±0.2 | 1.0±0.1 | 0.112 |
| | LV systolic EI | 1.6±0.8 | 1.0±0.1 | 0.058 | 1.5±0.4 | 1.0±0.0 | 0.002 |
| | sPAP (mmHg) | 70.5±29.7 | 35.8±6.9 | 0.003 | 79.2±31.8 | 34.1±10.8 | <0.001 |
| | TAPSE/sPAP ratio | 0.34±0.19 | 0.64±0.18 | 0.002 | 0.33±0.20 | 0.59±0.18 | 0.001 |
| Hemodynamics | RAP (mmHg) | 7.7±4.8 | 5.8±3.9 | 0.191 | 6.9±3.4 | 5.9±3.5 | 0.450 |
| | sPAP (mmHg) | 74.1±21.6 | 40.8±17.3 | <0.001 | 75.1±26.6 | 36.0±10.6 | <0.001 |
| | dPAP (mmHg) | 27.7±9.9 | 15.4±7.4 | 0.001 | 23.4±8.0 | 12.5±3.9 | <0.001 |
| | mPAP (mmHg) | 44.8±12.4 | 26.1±9.3 | <0.001 | 42.1±12.9 | 22.6±5.4 | <0.001 |
| | PCWP (mmHg) | 9.9±2.6 | 11.3±3.7 | 0.349 | 11.6±4.8 | 11.0±3.6 | 0.570 |
| | SvO$_2$ (%) | 61.4±9.0 | 69.1±6.1 | 0.023 | 67.0±6.8 | 71.4±4.7 | 0.015 |
| | CO (L/min) | 4.2±1.2 | 5.0±0.9 | 0.035 | 4.2±1.3 | 4.8±1.0 | 0.007 |
| | CI (L/min/m²) | 2.2±0.5 | 2.7±0.4 | 0.011 | 2.4±0.6 | 2.6±0.4 | 0.492 |
| | PVR (WU) | 9.8±4.6 | 3.0±1.3 | <0.001 | 9.0±5.4 | 2.9±1.9 | <0.001 |
| | PAC (mL/mmHg) | 1.4±0.8 | 3.2±1.8 | 0.001 | 1.5±1.0 | 3.5±2.1 | <0.001 |
| | HR (bpm) | 77.2±12.6 | 71.2±10.9 | 0.281 | 71.6±8.8 | 72.6±11.1 | 0.782 |

6MWD: 6-min walk distance; BSA: body surface area; BPA: balloon pulmonary angioplasty; EI – excentricity index; CO: cardiac output; CI: cardiac index; dPAP: diastolic pulmonary arterial pressure; HR: heart rate; LV: left ventricle; mPAP: mean pulmonary arterial pressure; PAC: pulmonary arterial compliance; PCWP: pulmonary capillary wedge pressure; PEA: pulmonary endarterectomy; PVR: pulmonary vascular resistance; RA: right atrium; RV FAC: right ventricular fractional area change; SvO$_2$: mixed venous oxygen saturation; sPAP: systolic pulmonary arterial pressure; TAPSE: tricuspid annular plane systolic excursion; WHO FC: World Heath Organization functional class.

In the BPA pathway, early reductions in mPAP and PVR were partly driven by pulmonary vasodilator therapy, with additional significant improvements after BPA (Fig 3), corresponding to mean patient-level reductions of 29% in mPAP and 51% in PVR, and group-level reductions of 32% and 54%, respectively. Hemodynamic benefits persisted at long-term follow-up in both pathways.

Fig 4 illustrates the proportion of patients exceeding predefined hemodynamic thresholds at follow-up. Despite sustained improvements from baseline, most patients had mPAP values above the normal limit of 20 mmHg at long-term follow-up. When patients with residual pulmonary hypertension at 6 months post-surgery who subsequently crossed over to BPA were also included, 23.1% in the BPA group and 27.8% in the PEA group had post-intervention mPAP values >30 mmHg, a threshold consistently associated with worse prognosis.

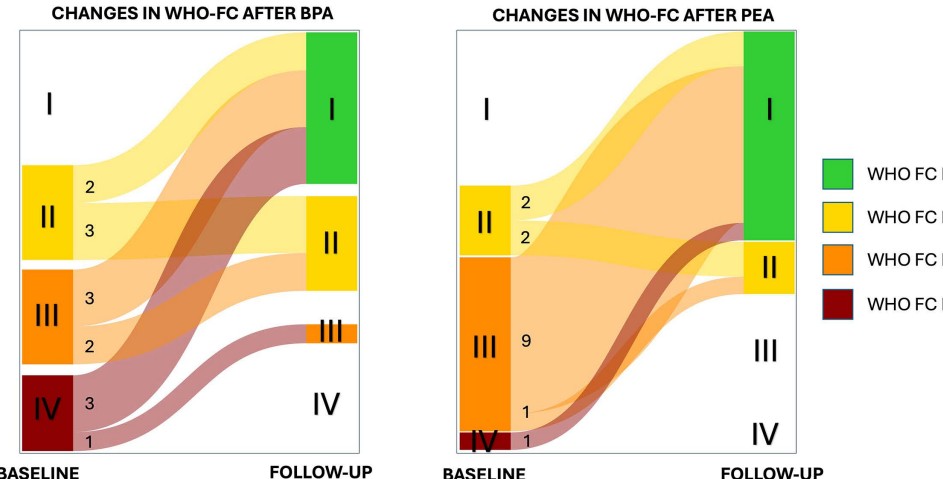

**Fig 2. Sankey diagram illustrating changes in WHO Functional Class (WHO FC) from baseline to follow-up in the BPA and PEA groups.** *Each vertical node represents a WHO FC category at baseline (left) and follow-up (right). The width of each flow is proportional to the number of patients transitioning between classes.*

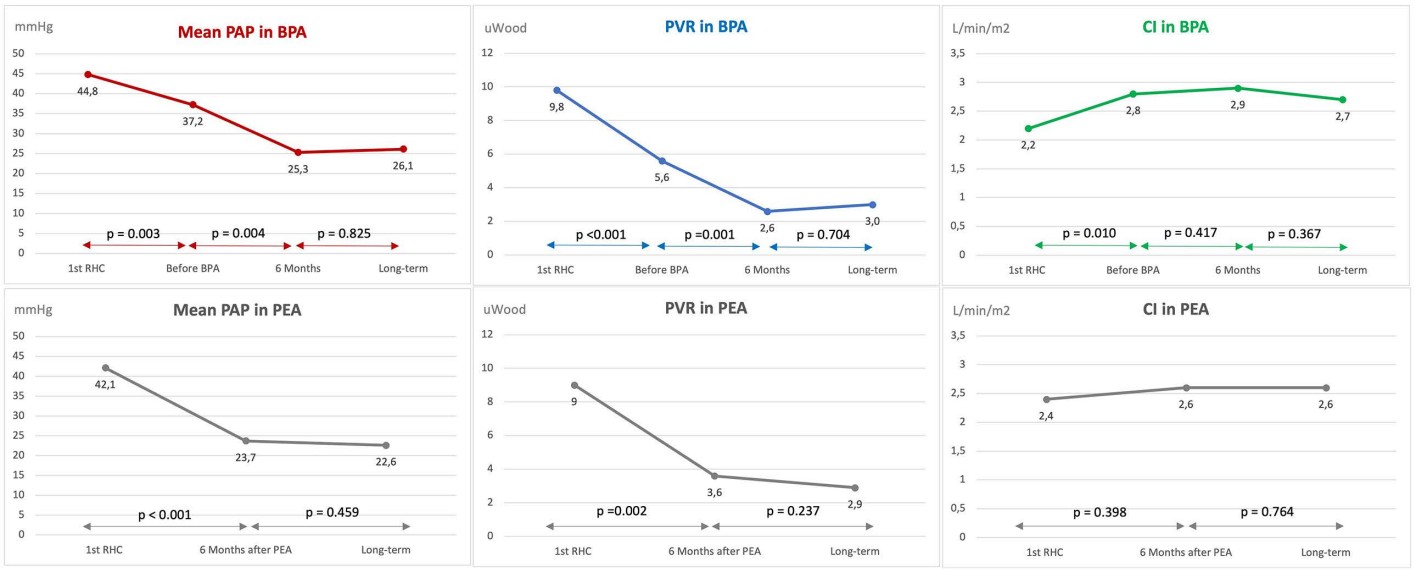

**Fig 3. Changes over time in mean pulmonary arterial pressure, pulmonary vascular resistance and cardiac index in both groups.** 1st RHC: first right heart catheterization; BPA: balloon pulmonary angioplasty; CI: cardiac index; PAP: pulmonary arterial pressure; PEA: pulmonary endarterectomy; PVR: pulmonary vascular resistance.

## Results in Exercise RHC at long-term

Ten patients in the BPA group and 12 patients in the PEA group underwent exercise RHC at long-term follow-up, with a median interval of 42 months (IQR 29–48) between BPA and exercise RHC and 51 months (IQR 38–62) between PEA and exercise RHC. Exercise duration ranged from 9 to 15 minutes, and pulmonary artery oxygen saturation at peak exercise was approximately 45%.

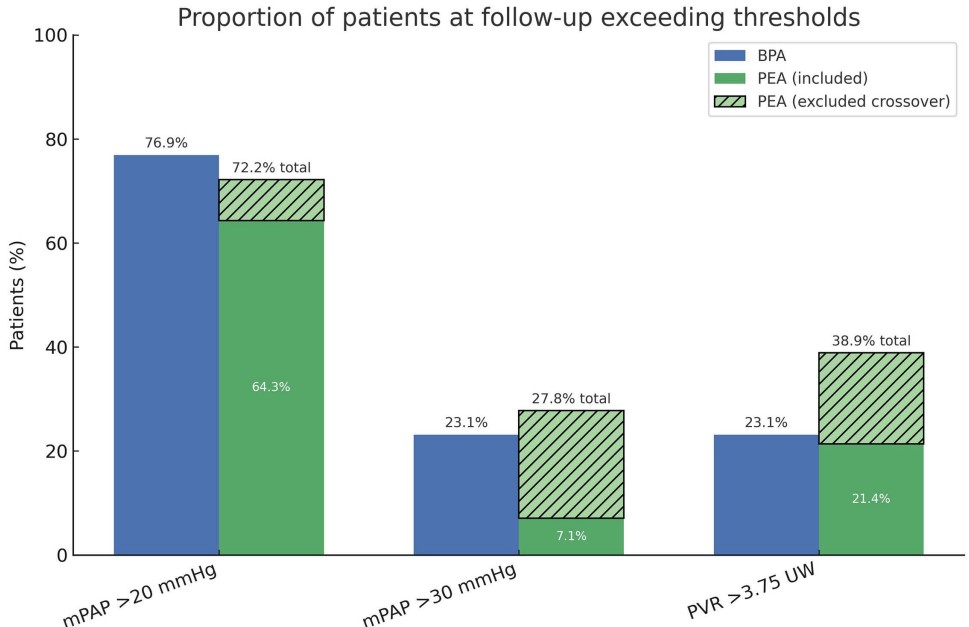

**Fig 4. Proportion of patients at follow-up exceeding predefined hemodynamic thresholds.** Bars show the percentage of patients in the BPA and PEA groups with mPAP>20 mmHg, mPAP>30 mmHg, and PVR>3.75 Wood units at follow-up. For the PEA group, four patients with mPAP>30 mmHg and PVR>3.75 WU at 6 months who subsequently crossed over to BPA were excluded from the main analysis and are represented as the hatched upper segment of the green bars.

Fig 5 represents individual mPAP/CO slopes following BPA and PEA.

At long-term follow-up, exercise RHC demonstrated elevated mPAP/CO slopes in patients from both treatment pathways. Mean mPAP/CO slope values were 7.0±5.6 mmHg/L/min after BPA and 4.0±2.3 mmHg/L/min after PEA. A high proportion of patients exhibited slopes exceeding 3.0 mmHg/L/min, indicating persistent abnormal pulmonary vascular responses to exercise despite intervention (Fig 6).

Two BPA patients and five PEA patients had a PCWP/CO slope >2 mmHg/L/min, consistent with a post-capillary contribution to exercise PH.

## Results in health-related quality of life (HRQOL)

At long-term follow-up, 13 BPA and 15 PEA patients completed the SF-36 quality of life questionnaire (Table 3), with a median interval of 51 months (IQR 37–60) after BPA and 55 months (IQR 39–75) after PEA. Physical health–related quality of life remained impaired, with PCS scores of 44.4±12.7 after BPA and 44.5±7.3 after PEA, both below the population norm of 50. In contrast, the MCS showed a more pronounced reduction in the BPA pathway, whereas mean values after PEA were close to the population norm.

## Exercise hemodynamics and quality of life according to post-procedural resting hemodynamic status and vasodilator therapy

To further explore the concept of disease resolution after PEA or BPA, exploratory analyses were performed at long-term follow-up according to post-procedural resting hemodynamic status and use of pulmonary vasodilator therapy.

At long-term follow-up, eight patients achieved normal resting pulmonary hemodynamics (mPAP<20 mmHg; 23.1% in the BPA group and 35.7% in the PEA group). Even in this subgroup, patients exhibited abnormal exercise

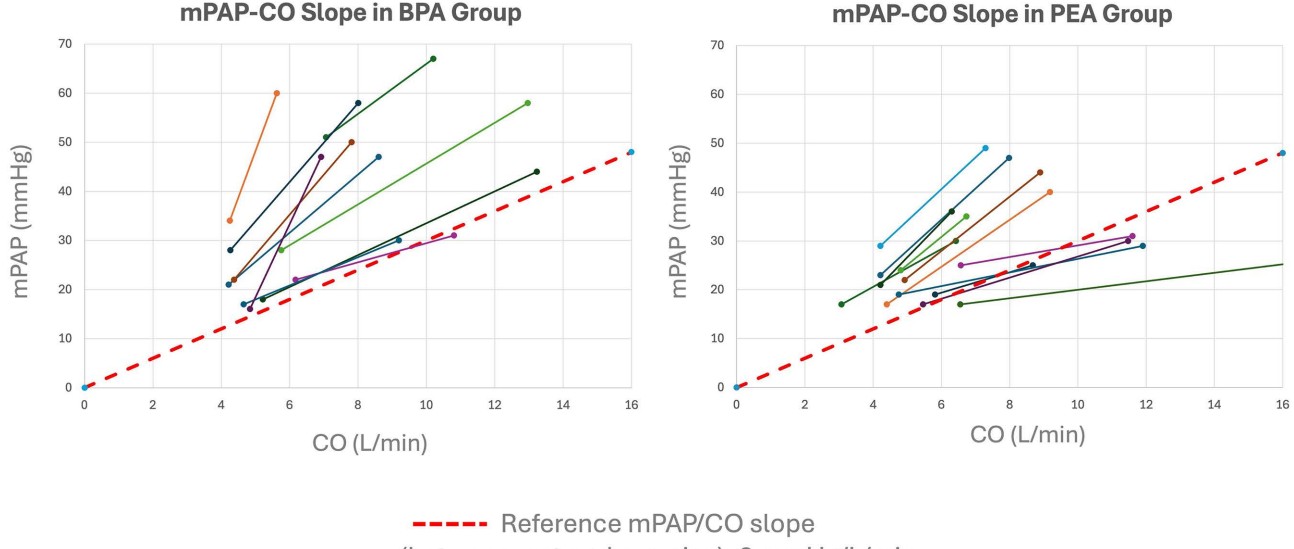

**Fig 5. Individual mPAP/CO slopes in patients treated with BPA (left) and PEA (right).** *Each line represents one patient, connecting resting and peak exercise measurements of mean pulmonary arterial pressure (mPAP) versus cardiac output (CO). The red dashed line represents the upper limit of normal for the mPAP/CO slope (3 mmHg/L/min).*

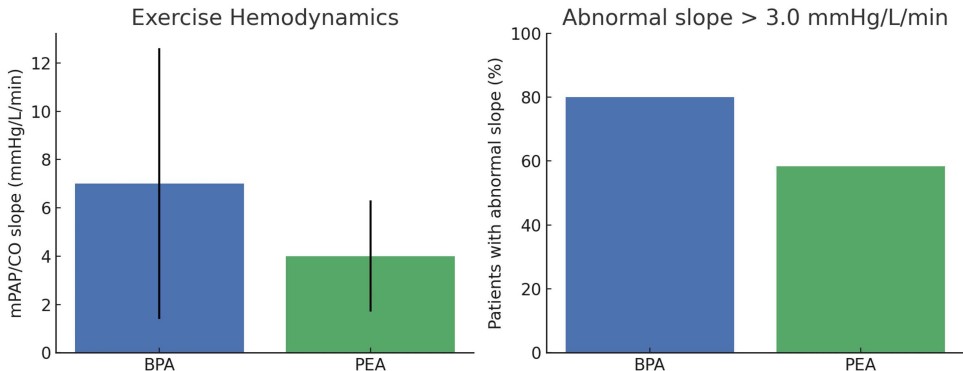

**Fig 6. Exercise hemodynamics in both BPA and PEA pathways.** *Left: Mean mPAP/CO slope; Right: proportion of patients with an abnormal slope (>3.0 mmHg/L/min).*

responses, with a mean mPAP/CO slope of 4.2 ± 4.5 mmHg/L/min, above the normal reference value. Patients with residual pulmonary hypertension had numerically steeper slopes (6.0 ± 4.2 mmHg/L/min), although the difference was not statistically significant (p = 0.378). Health-related quality of life did not differ between patients with normalized and impaired resting hemodynamics, both for PCS (42.5 ± 7.7 vs 46.9 ± 10.3, p = 0.302) and MCS scores (45.1 ± 14.4 vs 49.9 ± 13.4, p = 0.434).

At follow-up, 11 patients were receiving pulmonary vasodilator therapy (10 BPA, 1 PEA). These patients had more severe post-procedural resting hemodynamics, with higher mPAP (28.5 ± 9.8 vs 21.4 ± 3.6 mmHg, p = 0.040), whereas the mPAP/CO slope was not significantly different compared with patients off vasodilator therapy (6.2 ± 5.4 vs 4.9 ± 3.6 mmHg/L/min, p = 0.506).

**Table 3. Long-term HRQOL scores.**

| Domain | Follow-up BPA (N = 13) | Follow-up PEA (N = 15) |
|---|---|---|
| Physical Functioning (PF) | 42.1 ± 13.8 (66.2 ± 31.6) | 41.3 ± 12.6 (64.3 ± 28.8) |
| Role Physical (RP) | 43.5 ± 13.3 (59.6 ± 43.9) | 46.5 ± 13.0 (69.6 ± 42.9) |
| Bodily Pain (BP) | 46.2 ± 14.3 (66.2 ± 33.9) | 50.6 ± 10.9 (76.7 ± 25.8) |
| General Health (GH) | 39.9 ± 11.0 (50.8 ± 23.0) | 42.5 ± 10.9 (56.3 ± 22.9) |
| Vitality (VT) | 47.1 ± 12.2 (55.0 ± 25.4) | 51.1 ± 12.5 (63.3 ± 26.1) |
| Social Functioning (SF) | 42.1 ± 10.3 (65.4 ± 23.5) | 48.2 ± 11.5 (79.2 ± 26.2) |
| Role Emotional (RE) | 41.5 ± 12.3 (53.8 ± 39.8) | 46.9 ± 13.3 (71.4 ± 43.1) |
| Mental Health (MH) | 40.3 ± 15.3 (57.2 ± 27.5) | 45.8 ± 15.6 (67.2 ± 28.0) |
| Physical Component Summary (PCS) | 44.4 ± 12.7 | 44.5 ± 7.3 |
| Mental Component Summary (MCS) | 42.7 ± 14.3 | 49.9 ± 14.1 |

HRQOL: health-related quality of life

Values are presented as mean ± SD. *Values are presented as normalized T-scores (mean = 50, SD = 10); raw scores (0–100) are shown in parentheses.*

## Discussion

In this prospective single-center study with long-term follow-up, both PEA and BPA were associated with marked and sustained improvements in functional status and resting pulmonary hemodynamics in patients with CTEPH. However, despite these favorable resting changes, incomplete physiological recovery was common, with persistent abnormalities in exercise pulmonary vascular responses and impaired physical quality of life observed in most patients.

The magnitude of resting hemodynamic improvement in both groups is consistent with prior observational and controlled studies, further reinforcing the role of PEA as the standard of care and BPA as a valuable alternative for inoperable patients [15,25]. The BPA outcomes observed in this study are also aligned with the International Multicenter Prospective BPA Registry [17], which reported substantial improvements in clinical status and hemodynamics after a median of five sessions, including a 38% reduction in mPAP and 57% decrease in PVR, with complications occurring in 11.3% of sessions and 33.9% of patients, and no 30-day mortality. The safety profile of BPA in this cohort is therefore consistent with contemporary real-world experience, supporting the feasibility of this approach in expert centers.

These findings extend previous short-term reports from the same group [26], by demonstrating the durability of these improvements over a median follow-up exceeding three years.

Beyond resting hemodynamics, both interventions were associated with indices of right ventricular reverse remodeling. Improvements in RV function, reflected by increases in FAC and TAPSE/sPAP ratio, are consistent with prior imaging and echocardiographic studies showing substantial recovery of RV structure and function after PEA and [27], more recently, after BPA [28,29]. These changes likely contribute to the favorable long-term survival reported after both interventions [30], which in our cohort was comparable to previously published series [31,32].

A key contribution of this study is the long-term assessment of exercise hemodynamics using invasive exercise right heart catheterization. Prior work by Wiedenroth et al. [33] demonstrated that BPA improves hemodynamics at rest and during exercise at 6 months, including a reduction in the mPAP/CO slope (from 11.2 ± 25.6 to 7.7 ± 4.1 mmHg/L/min),

although values remained abnormal in many patients. In this cohort, abnormal exercise responses were likewise frequent at long-term follow-up after BPA, extending these observations beyond the early post-intervention phase.

These findings are also consistent with previous studies reporting persistent functional limitations despite hemodynamic improvement after BPA. Miura et al. [34] demonstrated that, in CTEPH patients with partial hemodynamic improvement after BPA, exercise tolerance and health-related quality of life frequently remain impaired despite favorable resting measurements. Similarly, Kikuchi et al. [35] showed that exercise intolerance often persists after successful BPA and is largely related to a limited increase in cardiac output during exertion, suggesting residual pulmonary vascular and microvascular dysfunction.

Data on exercise hemodynamics after PEA are scarce and largely limited to patients with chronic thromboembolic disease without resting pulmonary hypertension (CTED/CTEPD), a population that differs from our CTEPH cohort [36]. Although this study reports improvements in exercise hemodynamics after surgery, our findings in CTEPH patients show that abnormal mPAP/CO slopes frequently persist after PEA, indicating incomplete restoration of pulmonary vascular reserve despite marked improvements in resting hemodynamics.

These findings extend this literature by providing long-term invasive exercise hemodynamic data in both groups (BPA and PEA), highlighting persistent abnormalities in pulmonary vascular responses in both pathways despite substantial improvements in resting measurements. This underscores the importance of incorporating exercise testing into the routine follow-up of CTEPH patients, as reliance solely on resting hemodynamics may underestimate residual disease burden and functional limitation.

The persistence of abnormal exercise responses likely reflects a combination of residual pulmonary vascular disease burden and remodeling processes that may not be fully reversible with current therapies. Because complete anatomical revascularization was not systematically assessed, we cannot distinguish irreversible small-vessel vasculopathy from potentially incomplete reperfusion or residual treatable lesions; therefore, our findings should be interpreted as evidence of frequent incomplete physiological normalization in this cohort, rather than definitive proof of non-curability. An additional consideration when interpreting these findings is that both PEA and BPA are highly operator- and center-dependent pro-cedures. The magnitude of hemodynamic improvement and the likelihood of residual physiological abnormalities may vary according to procedural strategy, completeness of revascularization, institutional volume, and operator experience, par-ticularly given the recognized learning curve associated with BPA programs [17,37,38]. Similarly, outcomes after PEA are strongly influenced by institutional volume and team expertise, with improved results consistently reported in high-volume referral centers with dedicated multidisciplinary programs [10,39]. Therefore, post-interventional physiological outcomes should be interpreted within the context of technical and institutional factors, which may influence the extent of recovery observed.

Importantly, residual physiological abnormalities were mirrored by patient-reported outcomes. Physical health–related quality of life remained impaired in both groups at long-term follow-up, with PCS scores below population norms, despite hemodynamic improvement. Notably, patients treated with BPA achieved quality of life levels comparable to those after PEA, consistent with previous reports [40,41], although physical limitations frequently persisted [42]. Mental health–related quality of life, as reflected by the Mental Component Summary (MCS), was below the population norm of 50 in patients treated with BPA at long-term follow-up. This finding may relate to patient characteristics commonly associated with this treatment pathway, including older age and a higher burden of comorbid conditions such as chronic lung disease, which may contribute to greater psychological and functional impact over time.

In exploratory subgroup analyses, normalization of resting mPAP was not associated with better quality of life or normal exercise hemodynamics, underscoring the multifactorial determinants of symptoms and functional limitation in CTEPH.

Finally, patients receiving pulmonary vasodilator therapy at follow-up exhibited more severe residual resting hemody-namics, consistent with greater disease burden, yet without a significant difference in mPAP/CO slope. This supports the notion that improvements in resting load do not necessarily translate into normalization of dynamic pulmonary vascular responses during exercise.

## Limitations

This single-center study has a modest sample size, which limits generalizability and precludes robust inferential comparisons between treatment pathways. Accordingly, the analyses were descriptive in nature and focused on within-pathway changes and overall patterns observed after intervention. Larger multicenter studies will be needed to definitively compare BPA and PEA in terms of exercise hemodynamics and quality of life at long-term. In addition, not all patients underwent invasive exercise testing at follow-up, introducing potential selection bias. Nevertheless, the prospective design, extended follow-up duration, and use of invasive exercise hemodynamics represent important strengths of this study.

Health-related quality of life was assessed using the SF-36, a generic questionnaire rather than a pulmonary hypertension–specific patient-reported outcome measure. Although disease-specific instruments may be more sensitive to PH-related symptoms, the SF-36 was selected to allow comparison with normative population values, consistent with the study's focus on physiological recovery after intervention. The use of a generic measure may have underestimated residual symptom burden, and future studies should incorporate PH-specific questionnaires to provide a more comprehensive assessment of patient-centered outcomes.

## Conclusion

In this single-centre cohort, both pulmonary endarterectomy and balloon pulmonary angioplasty were associated with sustained improvements in resting pulmonary hemodynamics and functional status. However, complete physiological normalization was not achieved in a substantial proportion of patients, with persistent abnormalities in exercise hemodynamics and health-related quality of life at long-term follow-up. These findings highlight the frequent presence of residual pulmonary vascular dysfunction after intervention and underscore the importance of comprehensive long-term assessment beyond resting hemodynamics.

## Supporting information

**S1 Table. Patients managed with a hybrid PEA–BPA strategy.**
(DOCX)

## Aknowledgements

The authors used ChatGPT (OpenAI, version 5.2) to assist in drafting and editing some sections of this manuscript. All AI-generated text was reviewed, edited, and verified by the authors to ensure accuracy and compliance with academic standards.

## Author contributions

**Conceptualization:** Rita Calé, Filipa Ferreira, Mariana Martinho, Sofia Alegria, Daniel Caldeira.

**Data curation:** Rita Calé, Filipa Ferreira, Débora Repolho, Ana Rita Pereira, João Luz, Tiago Lobão, Patrícia Araújo, Sílvia Vitorino.

**Formal analysis:** Rita Calé, Filipa Ferreira, Mariana Martinho, Sofia Alegria, Débora Repolho, Ana Rita Pereira.

**Investigation:** Rita Calé.

**Methodology:** Rita Calé, Daniel Caldeira.

**Project administration:** Helder Pereira, Daniel Caldeira.

**Resources:** Helder Pereira.

**Supervision:** Helder Pereira, Daniel Caldeira.

**Validation:** Helder Pereira, Daniel Caldeira.

**Visualization:** João Luz, Tiago Lobão, Patrícia Araújo, Sílvia Vitorino.

**Writing – original draft:** Rita Calé.

**Writing – review & editing:** Rita Calé, Filipa Ferreira, Mariana Martinho, Sofia Alegria, Ana Rita Pereira, Daniel Caldeira.

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
