## [Decision Letter · Decision Letter 0]

15 Dec 2025

PONE-D-25-61676Is There a Cure for Chronic Thromboembolic Pulmonary Hypertension? Long-Term Functional and Exercise Hemodynamic Responses After PEA and BPAPLOS One?

Dear Dr. Calé,

Thank you for submitting your manuscript to PLOS ONE. After careful consideration, we feel that it has merit but does not fully meet PLOS ONE’s publication criteria as it currently stands. Therefore, we invite you to submit a revised version of the manuscript that addresses the points raised during the review process.

We look forward to receiving your revised manuscript.

Kind regards,

R. Jay Widmer

Academic Editor

PLOS One

**Journal Requirements:**

2. For studies involving third-party data, we encourage authors to share any data specific to their analyses that they can legally distribute. PLOS recognizes, however, that authors may be using third-party data they do not have the rights to share. When third-party data cannot be publicly shared, authors must provide all information necessary for interested researchers to apply to gain access to the data. (https://journals.plos.org/plosone/s/data-availability#loc-acceptable-data-access-restrictions)

**Additional Editor Comments:**

The reviewers have raised substantial concerns regarding this current version of the manuscript. Please address each comment individually in your revision letter. Thank you for this submission.

Reviewers' comments:

Reviewer's Responses to Questions

**Comments to the Author**

1. Is the manuscript technically sound, and do the data support the conclusions?

Reviewer #1: Yes

Reviewer #2: No

2. Has the statistical analysis been performed appropriately and rigorously?

Reviewer #1: Yes

Reviewer #2: Yes

3. Have the authors made all data underlying the findings in their manuscript fully available?

Reviewer #1: No

Reviewer #2: Yes

4. Is the manuscript presented in an intelligible fashion and written in standard English?

Reviewer #1: Yes

Reviewer #2: Yes

Reviewer #1: Cale and colleagues present long term data for 36 (29 in analysis) patients who underwent BPA or PEA in a single centre between 2017 and 2023. The manuscript is generally well written but the results section and conclusion was quite hard to read through due to the length of the sections. Parts of the discussion are not directly relevant to the manuscript and to help the reader should be removed.

Specific major comments

There were some patients who did not have PH post procedures - did these have better QoL or less steep mPAP-CO slopes? Did the patient on vasodilator therapy vs no vasodilator therapy have any difference in mPAP-CO slope? The number of patients may mean this may not be statistically significant but should be discussed because the title is talking about curing CTEPH patients.

The SF36 questionnaire is a general QoL measure and has been shown to be less effective PROM in PH patients compared to specific measures e.g. CAMPHOR, EMPHASIS-10, etc. Why was SF36 chosen. I understand at this stage it cannot be changed but this should be discussed.

Other authors have reported exercise haemodynamic outcomes and QoL after BPA and these should be discussed.

Specific minor comments

In multiples places the authors use the phase "mean systolic pressure" instead of "mean pulmonary artery pressure". Please correct.

anestesia (4th line of BPA section on page 8) needs to be corrected.

Reviewer #2: This single-centre case series reports the post-treatment status of patients with CTEPH after BPA or PEA. The authors employed a comprehensive assessment, including exercise right-heart catheterisation and patient-reported outcomes. The breadth of phenotyping is a key strength, and the dataset may be valuable in providing a detailed, real-world, single-centre depiction of residual physiological abnormalities after intervention.

The reviewer acknowledges the widely held clinical concept that CTEPH is not always fully “curable” even after BPA/PEA, in part because of concomitant small-vessel vasculopathy. Nevertheless, I do not consider that the present dataset, as currently analysed and presented, is sufficient to support such a general conclusion.

Accordingly, the current manuscript overreaches by concluding that CTEPH cannot be cured with BPA/PEA. At best, the data show that, in this centre/cohort, physiological normalisation is frequently incomplete.

Major comments

1. The conclusion that CTEPH cannot be cured is overstated.

This is a single-centre observational case series and cannot support a general statement that BPA/PEA cannot cure CTEPH. The manuscript should instead conclude that complete normalisation was not achieved in many patients in this cohort/centre.

2. “Not curable” requires a clear, operational definition and proof of complete revascularisation.

If the authors wish to argue “not curable,” they should define it a priori as persistent abnormalities in (i) resting haemodynamics, (ii) exercise pulmonary vascular response (mPAP/CO slope), and (iii) exercise tolerance despite complete anatomical revascularisation documented by contrast-enhanced CT pulmonary angiography and/or pulmonary angiography and/or V/Q scintigraphy.

At present, the manuscript does not demonstrate whether complete revascularisation was achieved after BPA or PEA, making it impossible to distinguish “non-curability” from potentially incomplete reperfusion therapy or residual treatable lesions.

3. Hybrid (PEA + BPA) cases should be included and described.

If this work is presented as a single-centre case series describing outcomes after intervention, hybrid management cases should not be excluded. Contemporary CTEPH care often involves BPA after PEA for residual PH (or vice versa), and excluding such patients limits clinical relevance and may bias the interpretation.

4. A comparative analysis of BPA versus PEA is not valid in this dataset.

Baseline characteristics, treatment pathways, and background therapy differ substantially between BPA and PEA patients (including pulmonary vasodilator use and process of care). The manuscript should avoid implying a head-to-head comparison and should frame the results as a descriptive case series of outcomes following two distinct treatment strategies.

5. The manuscript is overly long and should be reorganised.

The text is verbose, and descriptions of well-established examinations and procedural aspects of BPA/PEA can be substantially shortened. In contrast, it is appropriate to describe the exercise RHC protocol in detail, as protocols vary across centres and materially affect interpretation.

**Do you want your identity to be public for this peer review?** For information about this choice, including consent withdrawal, please see our Privacy Policy

Reviewer #1: No

Reviewer #2: No

---

## [Author Response · Author response to Decision Letter 1]

23 Jan 2026

Response to Reviewers

Reviewer #1: Cale and colleagues present long term data for 36 (29 in analysis) patients who underwent BPA or PEA in a single centre between 2017 and 2023. The manuscript is generally well written but the results section and conclusion was quite hard to read through due to the length of the sections. Parts of the discussion are not directly relevant to the manuscript and to help the reader should be removed.

Response:

We thank the reviewer for this helpful comment. In response, the Discussion was substantially revised to focus more directly on the primary objective of the study, namely the long-term assessment of resting and exercise hemodynamics and patient-reported outcomes after intervention. Sections that were less directly relevant to this aim were removed or condensed. In addition, the Results section was streamlined to improve readability. As a result of these revisions, the overall word count of the manuscript was reduced, and the flow and clarity of the Results and Discussion sections were improved.

Specific major comments

There were some patients who did not have PH post procedures - did these have better QoL or less steep mPAP-CO slopes? Did the patient on vasodilator therapy vs no vasodilator therapy have any difference in mPAP-CO slope? The number of patients may mean this may not be statistically significant but should be discussed because the title is talking about curing CTEPH patients.

Response:

We thank the reviewer for this insightful comment, which addresses the core concept of “cure” explored in our manuscript. In response, we performed additional exploratory analyses focusing on patients with normalized resting hemodynamics at follow-up and on the impact of pulmonary vasodilator therapy.

At long-term follow-up, eight patients achieved normal resting pulmonary hemodynamics (mPAP <20 mmHg; 23.1% in the BPA group and 35.7% in the PEA group). Despite normalization of resting mPAP, these patients still demonstrated abnormal exercise hemodynamics, with a mean mPAP/CO slope of 4.2±4.5 mmHg/L/min, exceeding the normal reference value of 3.0 mmHg/L/min. Patients with residual pulmonary hypertension exhibited numerically steeper slopes (6.0±4.2 mmHg/L/min), although the difference was not statistically significant (p=0.378). Importantly, no significant differences in health-related quality of life were observed between patients with normalized versus impaired resting hemodynamics, both for PCS (42.5±7.7 vs 46.9±10.3, p=0.302) and MCS scores (45.1±14.4 vs 49.9±13.4, p=0.434).

We also assessed the effect of pulmonary vasodilator therapy at follow-up. Eleven patients were receiving vasodilators (10 BPA, 1 PEA) and presented with more severe resting hemodynamics compared with those off therapy (mPAP 28.5±9.8 vs 21.4±3.6 mmHg, p=0.040). However, the mPAP/CO slope did not differ significantly between patients on versus off vasodilator therapy (6.2±5.4 vs 4.9±3.6 mmHg/L/min, p=0.506).

These findings have now been incorporated into the Results and Discussion sections. They highlight that normalization of resting pulmonary hemodynamics does not necessarily translate into normal exercise pulmonary vascular responses or improved quality of life, reinforcing our conclusion that CTEPH should be regarded as a chronic condition rather than a definitively curable disease, even after successful PEA or BPA.

The SF36 questionnaire is a general QoL measure and has been shown to be less effective PROM in PH patients compared to specific measures e.g. CAMPHOR, EMPHASIS-10, etc. Why was SF36 chosen. I understand at this stage it cannot be changed but this should be discussed.

Response:

The SF-36 was selected primarily because it is a widely validated, generic health-related quality of life instrument that allows comparison with normative population values and across different disease states. This was particularly relevant to our study, which aimed to explore the concept of “disease resolution” after PEA or BPA, including whether patients approach population-based norms after intervention.

In addition, the use of disease-specific PROMs such as CAMPHOR was not feasible, as this instrument requires licensing and associated costs, and the present study was conducted without dedicated external funding. At the time of study design, EMPHASIS-10 was not routinely implemented in our center’s clinical practice.

We fully acknowledge that disease-specific questionnaires may be more sensitive to subtle symptoms and functional limitations in pulmonary hypertension. We have now explicitly addressed this limitation in the Discussion section and recognize that the use of a generic instrument may have underestimated the burden of residual symptoms. Future studies should incorporate PH-specific PROMs to better capture patient-centered outcomes after PEA or BPA.

Other authors have reported exercise haemodynamic outcomes and QoL after BPA and these should be discussed.

Response:

We thank the reviewer for this important suggestion. In response, we have substantially revised and expanded the Discussion to incorporate evidence from previous studies addressing exercise hemodynamic outcomes, quality of life, and resting hemodynamics after BPA and PEA.

Specifically, we have contextualized our findings by comparing the magnitude of resting hemodynamic improvement after BPA with data from the International BPA Registry. We have also incorporated studies by Wiedenroth et al. and Kikuchi et al., which evaluated exercise right heart catheterization after BPA and demonstrated persistent abnormalities in exercise hemodynamics despite improvements at rest. In addition, we have discussed the study by Guth et al., which assessed exercise hemodynamics after PEA in patients with chronic thromboembolic disease, highlighting differences in study populations and outcomes.

Finally, we have included discussion of a PLOS ONE study reporting persistent exercise intolerance and impaired quality of life in CTEPH patients with partial hemodynamic improvement after BPA, which closely aligns with our observations. Together, these additions provide a more comprehensive and balanced discussion of existing literature on exercise hemodynamics and patient-reported outcomes after intervention and better position our findings within the current evidence base.

Specific minor comments

In multiples places the authors use the phrase "mean systolic pressure" instead of "mean pulmonary artery pressure". Please correct.

Response:

We thank the reviewer for this comment. We have now standardized the terminology throughout the manuscript, using the terms systolic, diastolic, and mean pulmonary arterial pressure consistently, abbreviated as sPAP, dPAP, and mPAP, respectively.

anestesia (4th line of BPA section on page 8) needs to be corrected.

Response:

We thank the reviewer for noting this. The spelling has been corrected from “anestesia” to “anesthesia.”

Reviewer #2: This single-centre case series reports the post-treatment status of patients with CTEPH after BPA or PEA. The authors employed a comprehensive assessment, including exercise right-heart catheterisation and patient-reported outcomes. The breadth of phenotyping is a key strength, and the dataset may be valuable in providing a detailed, real-world, single-centre depiction of residual physiological abnormalities after intervention.

The reviewer acknowledges the widely held clinical concept that CTEPH is not always fully “curable” even after BPA/PEA, in part because of concomitant small-vessel vasculopathy. Nevertheless, I do not consider that the present dataset, as currently analysed and presented, is sufficient to support such a general conclusion.

Accordingly, the current manuscript overreaches by concluding that CTEPH cannot be cured with BPA/PEA. At best, the data show that, in this centre/cohort, physiological normalisation is frequently incomplete.

Major comments

1. The conclusion that CTEPH cannot be cured is overstated.

This is a single-centre observational case series and cannot support a general statement that BPA/PEA cannot cure CTEPH. The manuscript should instead conclude that complete normalisation was not achieved in many patients in this cohort/centre.

2. “Not curable” requires a clear, operational definition and proof of complete revascularisation.

If the authors wish to argue “not curable,” they should define it a priori as persistent abnormalities in (i) resting haemodynamics, (ii) exercise pulmonary vascular response (mPAP/CO slope), and (iii) exercise tolerance despite complete anatomical revascularisation documented by contrast-enhanced CT pulmonary angiography and/or pulmonary angiography and/or V/Q scintigraphy.

At present, the manuscript does not demonstrate whether complete revascularisation was achieved after BPA or PEA, making it impossible to distinguish “non-curability” from potentially incomplete reperfusion therapy or residual treatable lesions.

Response to question 1 and 2:

We thank the reviewer for this thoughtful and constructive critique, which we fully acknowledge. We agree that, given the single-centre observational nature of our study, our data cannot support a generalized statement that CTEPH is not curable after BPA or PEA.

We have therefore revised the manuscript to avoid overgeneralization and to more accurately reflect the scope of our findings. Specifically, we have reframed our conclusions to state that, in this cohort and at this centre, complete physiological normalization was not achieved in a substantial proportion of patients despite intervention, rather than asserting that CTEPH is categorically not curable.

We also agree that the term “curable” requires a clear operational definition. In the revised manuscript, we no longer claim non-curability per se, but instead focus on persistent abnormalities in exercise hemodynamics and patient-reported outcomes observed at long-term follow-up. Importantly, we acknowledge that our study did not systematically assess complete anatomical revascularisation after BPA or PEA using imaging modalities such as contrast-enhanced CT pulmonary angiography, pulmonary angiography, or V/Q scintigraphy. As such, we cannot distinguish residual physiological abnormalities due to irreversible small-vessel disease from potentially incomplete reperfusion or residual treatable lesions.

These limitations are now explicitly acknowledged in the Discussion. In addition, we have revised the title and the conclusions of both the abstract and the main manuscript to avoid implying definitive non-curability, and instead to emphasize the frequent persistence of physiological abnormalities after intervention, which represents the central and well-supported finding of our study.

3. Hybrid (PEA + BPA) cases should be included and described.

If this work is presented as a single-centre case series describing outcomes after intervention, hybrid management cases should not be excluded. Contemporary CTEPH care often involves BPA after PEA for residual PH (or vice versa), and excluding such patients limits clinical relevance and may bias the interpretation.

Response:

We thank the reviewer for this important point and fully agree that hybrid management is increasingly relevant in contemporary CTEPH care. Given the heterogeneity in treatment sequence and timing, patients undergoing hybrid management were not included in the main descriptive analysis of isolated treatment pathways. In response to the reviewer’s suggestion, we have now included a separate descriptive summary of hybrid cases in the Supplementary Material, detailing their clinical characteristics and hemodynamic outcomes. This approach has been clarified in the Methods, and a reference to the supplementary table has been added in the main manuscript.

4. A comparative analysis of BPA versus PEA is not valid in this dataset. Baseline characteristics, treatment pathways, and background therapy differ substantially between BPA and PEA patients (including pulmonary vasodilator use and process of care). The manuscript should avoid implying a head-to-head comparison and should frame the results as a descriptive case series of outcomes following two distinct treatment strategies.

Response:

We thank the reviewer for this important comment. Given the observational design and the distinct treatment pathways leading to BPA and PEA, we agree that a valid head-to-head comparison between strategies is not appropriate. Accordingly, the manuscript was revised to adopt a purely descriptive analytical framework. All tables were reviewed and revised to remove between-strategy p values, and baseline characteristics are now presented for descriptive purposes only. The Abstract, Statistical Analysis, and Results sections were also updated to ensure that no direct comparisons between treatment pathways are implied and that findings are reported as within-pathway changes and overall patterns following intervention. As no formal inferential comparisons are now presented, bootstrap resampling was removed, as it was no longer methodologically required.

5. The manuscript is overly long and should be reorganised.

The text is verbose, and descriptions of well-established examinations and procedural aspects of BPA/PEA can be substantially shortened. In contrast, it is appropriate to describe the exercise RHC protocol in detail, as protocols vary across centres and materially affect interpretation.

Response:

We thank the reviewer for this constructive comment. In response, the manuscript was substantially shortened and reorganised. The BPA protocol description was shortened, with detailed procedural aspects referenced to a previous publication by our group. Similarly, the description of the PEA procedure was condensed to include only the essential methodological aspects, as surgical techniques are well established and extensively described in the literature. In contrast, the description of the exercise right heart catheterization protocol was clarified, given its methodological importance and variability across centers.

the Results section was also substantially condensed to focus on the key findings of the analysis. Descriptions of secondary or less central results were shortened, allowing greater emphasis on the main outcomes of interest, particularly long-term resting and exercise hemodynamics and patient-reported quality of life.

These revisions substantially reduced the manuscript’s length while improving overall clarity and flow.

---

## [Decision Letter · Decision Letter 1]

8 Feb 2026

Dear Dr. Calé,

Thank you for submitting your manuscript to PLOS ONE. After careful consideration, we feel that it has merit but does not fully meet PLOS ONE’s publication criteria as it currently stands. Therefore, we invite you to submit a revised version of the manuscript that addresses the points raised during the review process.

We look forward to receiving your revised manuscript.

Kind regards,

R. Jay Widmer

Academic Editor

PLOS One

**Journal Requirements:**

**Additional Editor Comments:**

Overall the manuscript is markedly improved. The reviewers have some minor comments to address in your revisions prior to acceptance. We thank you for your substantial work and dedication to the revised version.

Reviewers' comments:

Reviewer's Responses to Questions

**Comments to the Author**

Reviewer #1: All comments have been addressed

Reviewer #2: All comments have been addressed

2. Is the manuscript technically sound, and do the data support the conclusions?

Reviewer #1: Yes

Reviewer #2: Yes

3. Has the statistical analysis been performed appropriately and rigorously?

Reviewer #1: Yes

Reviewer #2: Yes

4. Have the authors made all data underlying the findings in their manuscript fully available?

Reviewer #1: Yes

Reviewer #2: Yes

5. Is the manuscript presented in an intelligible fashion and written in standard English?

Reviewer #1: Yes

Reviewer #2: Yes

Reviewer #1: (No Response)

Reviewer #2: General comments

The revised manuscript is substantially improved compared with the previous version. The tone and conclusions are now better aligned with the descriptive, single-centre observational nature of the dataset. I also appreciate the clearer framing of BPA and PEA as distinct treatment pathways rather than as a head-to-head comparative study, as well as the overall streamlining of the text. The inclusion and description of hybrid cases in the supplementary material is a welcome addition.

Overall, the manuscript is now much closer to being suitable for publication. I have one remaining major point and a small number of minor comments.

Major comment

Operator- and centre-dependence should be more explicitly acknowledged as a determinant of post-interventional improvement

While the Discussion appropriately avoids over-attributing residual abnormalities to fixed “non-curability,” it still under-emphasizes an important practical consideration: both PEA and BPA are highly operator- and centre-dependent interventions. The magnitude of improvement—and the likelihood of residual physiological abnormalities—can vary meaningfully with procedural strategy, completeness of revascularisation, institutional volume, and operator experience/learning curve.

Minor comments

1. “mean systolic pulmonary arterial pressure” in Figure 4 should be corrected.

**Do you want your identity to be public for this peer review?** For information about this choice, including consent withdrawal, please see our Privacy Policy

Reviewer #1: No

Reviewer #2: **Yes:** Kazuya Hosokawa, MD, PhD.

Associate professor, Advanced open innovation center, Kyushu University, Fukuoka, Japan

---

## [Author Response · Author response to Decision Letter 2]

10 Feb 2026

We thank the reviewers for the continued careful evaluation of our manuscript. We agree that operator and center experience represent key determinants of post-interventional outcomes in CTEPH. Accordingly, we have incorporated this consideration explicitly into the Discussion. To support this statement, we added three additional references addressing the learning curve and institutional expertise associated with BPA, as well as the impact of center volume and multidisciplinary experience on outcomes after PEA.

We also revised the Methods section to explicitly describe the institutional and operator experience for both treatment pathways. Specifically, we clarified that PEA procedures were predominantly performed in high-volume referral centers with established expertise in CTEPH management, and that BPA procedures were conducted by the same two dedicated operators, with the initial cases proctored by an experienced high-volume BPA operator. We believe this addition appropriately contextualizes the post-interventional outcomes and acknowledges the potential influence of operator and center experience.

---

## [Decision Letter · Decision Letter 2]

18 Feb 2026

Residual Physiological Abnormalities After Pulmonary Endarterectomy and Balloon Pulmonary Angioplasty in CTEPH

PONE-D-25-61676R2

Dear Dr. Calé,

We’re pleased to inform you that your manuscript has been judged scientifically suitable for publication and will be formally accepted for publication once it meets all outstanding technical requirements.

Kind regards,

R. Jay Widmer

Academic Editor

PLOS One

Additional Editor Comments (optional):

We thank the authors for their careful attention to each comment and their thoughtful revisions.

Reviewers' comments:

Reviewer's Responses to Questions

**Comments to the Author**

Reviewer #2: All comments have been addressed

2. Is the manuscript technically sound, and do the data support the conclusions?

Reviewer #2: Yes

3. Has the statistical analysis been performed appropriately and rigorously?

Reviewer #2: Yes

4. Have the authors made all data underlying the findings in their manuscript fully available?

Reviewer #2: Yes

5. Is the manuscript presented in an intelligible fashion and written in standard English?

Reviewer #2: Yes

Reviewer #2: The revised manuscript is substantially improved compared with the previous version. The authors have responded constructively to prior feedback, and the overall tone and conclusions are now appropriately aligned with the descriptive, single-centre observational nature of the dataset.

Overall, I consider that the authors have provided an adequate and appropriate response to the major methodological and conceptual concerns previously raised.

**Do you want your identity to be public for this peer review?** For information about this choice, including consent withdrawal, please see our Privacy Policy

Reviewer #2: No

---

## [Editor Report · Acceptance letter]

PONE-D-25-61676R2

PLOS One

Dear Dr. Calé,

I'm pleased to inform you that your manuscript has been deemed suitable for publication in PLOS One. Congratulations! Your manuscript is now being handed over to our production team.

Kind regards,

on behalf of

Dr. R. Jay Widmer

Academic Editor

PLOS One